# Antifungal Activity of 2-Allylphenol Derivatives on the *Botrytis cinerea* Strain: Assessment of Possible Action Mechanism

**DOI:** 10.3390/ijms24076530

**Published:** 2023-03-31

**Authors:** Andres F. Olea, Julia Rubio, Claudia Sedan, Denisse Carvajal, Maria Nuñez, Luis Espinoza, Ligia Llovera, Gerard Nuñez, Lautaro Taborga, Héctor Carrasco

**Affiliations:** 1Grupo QBAB, Instituto de Ciencias Químicas Aplicadas, Facultad de Ingeniería, Universidad Autónoma de Chile, San Miguel, Santiago 8900000, Chile; 2Departamento de Química, Universidad Técnica Federico Santa María, Avenida España 1680, Valparaíso 2340000, Chile

**Keywords:** *Botrytis cinerea*, antifungal activity, 2-allylphenol, alternative oxidase enzyme, AOX inhibitors

## Abstract

*Botrytis cinerea* is a phytopathogenic fungus that causes serious damage to the agricultural industry by infecting various important crops. 2-allylphenol has been used in China as a fungicide for more than a decade, and it has been shown that is a respiration inhibitor. A series of derivatives of 2-allylphenol were synthesized and their activity against *B. cinerea* was evaluated by measuring mycelial growth inhibition. Results indicate that small changes in the chemical structure or the addition of substituent groups in the aromatic ring induce important variations in activity. For example, changing the hydroxyl group by methoxy or acetyl groups produces dramatic increases in mycelial growth inhibition, i.e., the IC_50_ value of 2-allylphenol decreases from 68 to 2 and 1 μg mL^−1^. In addition, it was found that the most active derivatives induce the inhibition of *Bcaox* expression in the early stages of *B. cinerea* conidia germination. This gene is associated with the activation of the alternative oxidase enzyme (AOX), which allows fungus respiration to continue in the presence of respiratory inhibitors. Thus, it seems that 2-allylphenol derivatives can inhibit the normal and alternative respiratory pathway of *B. cinerea*. Therefore, we believe that these compounds are a very attractive platform for the development of antifungal agents against *B. cinerea*.

## 1. Introduction

The fungus *Botrytis cinerea*, responsible for the so-called gray mold disease, is a phytopathogen that causes serious damage to the agricultural industry, mainly due to its ability to infect various important crops, as well as its tendency to generate resistance to synthetic fungicides used to control this infection [1,2,3,4]. Multidrug resistance (MDR) is associated with loss of fungus sensitivity to chemically unrelated compounds such as phenylpyrroles, anilinopyrimidines, and dicarboximide [5]. It has been found that these resistant isolates can overexpress membrane transport proteins that allow the elimination of toxic compounds outside of the fungus [6,7]. Thus, the emergence of MDR-resistant isolates has become a major global issue due to the high appearance frequency of mutants [8,9,10].

These facts have prompted the search for new crop protection agents that are more effective against *B. cinerea* and, simultaneously, nontoxic and friendly to the environment. In this line, numerous studies are showing that medicinal plants are a good source of new bioactive molecules. These natural products are mainly secondary metabolites that have been isolated from plants and then chemically modified [11,12,13]. Following this path, during the last decades, there has been a strong effort to apply natural products, essential oils, and plant extracts to control plant diseases [13,14,15,16,17,18,19]. Natural products with antifungal activity can be found in plant extracts or essential oils, and are mainly phenolic compounds, such as phenylpropanoids, coumarins, flavonoids, or terpenes [18,20]. Some examples of natural antifungal products and their hemisynthetic derivatives with antifungal activity against *B. cinerea* are geranylated phenols [21,22,23], terpenes [24,25,26,27], and phenylpropanoids [28,29,30,31].

Phenolic compounds are synthesized by plants through the phenylpropanoid pathway and form part of the plant protection mechanisms against abiotic stress and induced microorganism diseases [32]. Due to their interesting biological activities, some of these molecules have been tested for potential effects on human health (Figure 1). For instance, eugenol (**1**) and chlorogenic acid (**2**) exhibit antifungal, antioxidant, and anticancer properties [33,34,35,36,37]. It has also been proposed that (**2**) possesses cognitive and neuroprotection properties [38], and can also help regulate glucose and lipid metabolism [39].

The antifungal effects of these compounds on phytopathogenic fungi have also been assessed [30,31,40]. Eugenol and its hemisynthetic derivatives present activity on the mycelial growth of a virulent and multiresistant isolate of *B. cinerea* (PN2), with IC_50_ values that vary between 31 and 95 μg mL^−1^ [27]. These results indicate that some eugenol derivatives have a much higher antifungal activity or, at least, similar to that of BC-1000 (Chemie SA), a natural fungicide used for the control of *B. cinerea*. Additionally, it has been shown that the antifungal activity and chemical structure of these compounds is tightly related. Structure–activity analysis suggests that these compounds could exert their activity on fungi through two different routes, one of them would involve accumulation in the fungal membrane and the other through chemical reactions with unsaturated chains or reduction mediated by enzymes. The first mechanism would be associated with the lipophilic character of these molecules, while the second is due to the presence of active electron-attracting groups in the aromatic ring. This action leads to the disruption of the fungus membranes and ROS production [40]. Similarly, it has been demonstrated that chlorogenic acid inhibits the spore germination of *B. cinerea* by inducing membrane permeabilization in fungal spores, and at the same time, efficiently inhibits hyphal growth [31]. Additionally, it has been shown that the concentration of (**2**) and the activity of enzymes involved in the phenylpropanoid are increased in apple fruits infected by *B. cinerea* [41].

On the other hand, it has been shown that 2-allylphenol (**3**), a chemical compound used as a fungicide in China, is more effective against *B. cinerea* than iprodione [42]. Results indicate that (**3**) is a respiration inhibitor and, therefore, its effect is mainly fungistatic rather than fungicidal [43]. Additionally, the antifungal activity of (**3**) and its biotransformation products, 2-(2-hydroxypropyl) phenol (**4**) and 2-(3-hydroxypropyl) phenol (**5**), were determined [44]. The results show that only 2-(2-hydroxypropyl) phenol effectively inhibits the mycelial growth of *Rhizoctonia cerealis*, *Pythium aphanidermatum*, *Valsa mali*, and *B. cinerea*, with IC_50_ values ranging between 1.0 to 23.5 μg mL^−1^, which are lower than IC_50_ obtained for (**3**), i.e., 8.2 to 48.8 μg mL^−1^ [44].

Recently, 2-allylphenol and some of its derivatives were tested against *Phytophthora cinnamomi*. Interestingly, the antifungal activity is enhanced in those compounds where a nitro group is attached in the para position relative to the hydroxyl group. Additionally, several of these 2-allylphenol derivatives exhibit antifungal activities similar to or better than metalaxyl^®^, a commercial fungicide commonly used to control *P. cinnamomi* [28].

Thus, in this work, a series of derivatives of (**3**) were synthesized and their activity against *B. cinerea* was evaluated. The results are discussed in terms of a structure–activity relationship and then a probable mechanism of action is presented.

## 2. Results and Discussion

### 2.1. Synthesis of 2-Allylphenol Derivatives

Derivatives of 2-allylphenol were obtained by either attaching electron-attracting groups (-NO_2_) to the aromatic ring in ortho and para positions, relative to the hydroxyl group, or converting -OH into methoxy or acetyl groups. The chemical structures of all tested compounds are shown in Figure 2.

All derivatives, except 1-allyl-2-methoxy-5-nitrobenzene (**11**), have been previously synthesized and characterized by spectroscopic methods [45]. Compound (**11**) was obtained with an 89% yield by reacting (**7**) with dimethyl sulfate.

### 2.2. Evaluation of Antifungal Activity of 2-Allylphenol Derivatives on Botrytis cinerea

Antifungal activities of 2-allylphenol and its derivatives against *B. cinerea* were evaluated in vitro by measuring mycelial growth inhibition on PN2, a multiresistant isolate obtained from cherry fruits [24,30]. Figure 3 shows some typical results of radial growth measurements for this strain in the absence and presence of 2-allylphenol derivatives (**9**, **11**, and **13**) at different concentrations. The results indicate that all tested compounds affect the mycelial growth of *B. cinerea* in a concentration-dependent manner.

Growth percentage inhibition (%I) was calculated for each concentration, and the IC_50_ values (concentration causing 50% inhibition of mycelial growth) were obtained from fitting %I versus compound curves to a dose-response equation (see Section 3). IC_50_ values obtained for all assayed compounds are given in Table 1.

Results listed in Table 1 indicate that the IC_50_ varies from 1.00 to 136 μg mL^−1^, and the value obtained for 2-allylphenol (68.0 μg mL^−1^) is in the same range as that previously reported (48.8 μg mL^−1^) for the inhibition of B05.10, a nonresistant strain of *B. cinerea* [43,44]. Thus, the data indicate that growth inhibition of *B. cinerea*, evaluated as IC_50_ values, is clearly dependent on the chemical structure of 2-allylphenol derivatives.

The most striking changes in the activity of 2-allylphenol are shown by derivatives (**9**) and (**13**), in which the -OH group has been methoxylated or acetylated, respectively. Thus, compounds with no hydroxyl group exhibit higher antifungal activities (lower IC_50_ values) than (**3**), becoming very promising agents for the control of *B. cinerea*. Interestingly, the attachment of nitro groups induces different effects on the activity depending on their location in the aromatic ring. For example, for compounds (**3**), (**9**), and (**13**), a nitro group in the *ortho* position with respect to -OH induces a strong decrease in activity, whereas in the *para* position produces an increase, decrease, or no effect on the activity of (**3**), (**9**), and (**13**), respectively.

Gong et al. established that 2-allylphenol inhibits the mycelial growth of *B. cinerea* by inducing alternative respiration and depletion of ATP levels in the fungus [43]. These effects were attributed to electron-flow blocking by (**3**) in the cytochrome pathway, in which ubiquinol is oxidized to ubiquinone (coenzyme Q10). Ubiquinone is one of the most important and versatile of all electron transport components, having the ability to carry either one or two electrons and one or two protons. This molecule is formed by a water-soluble quinone and a long isoprenoid chain [46]. Considering the similitude of the chemical structures of ubiquinol and (**3**), it is likely that both compounds may be bound to the same receptors. Consequently, an alternative respiration pathway is induced. Additionally, all derivatives tested in this work maintain the chemical pattern of ubiquinol, i.e., an aromatic ring with an isoprenoid chain in the ortho position to a phenol, methoxy, or acetyl function. This feature ensures that all of them will affect *B. cinerea* following a similar mechanism. In the case of the most active derivatives, (**13**) and (**9**), the polar hydroxyl group has been changed by a methoxy or acetyl group, respectively (see Figure 2). Therefore, this increase in activity can be attributed to stronger binding to the receptor of these molecules due to a decrease in polarity. On the other hand, the attachment of the nitro group, a very strong electron acceptor, brings about different effects on activity depending on the parent compound and its position in the aromatic ring. For example, derivatives (**6**) and (**7**) differ only in the position of the nitro group on the aromatic ring, and their activities are completely different, as referred to in the parent compound (**3**). Derivative (**6**)**,** with a nitro group in the *ortho* position with respect to the hydroxyl, exhibits a decrease in activity, whereas the opposite effect is observed for derivative (**7**) in which the nitro group is in the *para* position. (Compare the IC_50_ values of (**3**) and (**6**); (**3**) and (**7**)). Similar effects are obtained for substitution with nitro groups in derivatives (**9**) and (**13**). The decrease in activity observed for derivatives with a nitro group in the *ortho* position could be explained in terms of the interaction of the nitro group with hydroxyl, methoxy, or acetyl groups in (**6**), (**10**), and (**14**), respectively, which makes the binding of these derivatives to the receptor weaker, and, consequently, the inhibition activity decreases (Compare the IC_50_ values of (**3**) and (**6**); (**9**) and (**10**); (**13**) and (**14**)).

### 2.3. Effect of 2-Allylphenol Derivatives on Gene Expression

It is well established that the respiratory chain of fungi is an effective target for fungicides designed to control fungal diseases in food crops [47,48,49]. However, it is also known that inhibition of the respiratory system activates the alternative oxidase enzyme (AOX), which participates in fungal mitochondrial oxidative phosphorylation, and, therefore, allows fungus respiration to continue despite the presence of inhibitors [50]. Thus, AOX activation limits the efficiency of respiratory inhibitors by providing a pathway by which the fungus recovers the ATP level necessary for its metabolic activity [50,51], thus playing a protective role against oxygen stress [52].

On the other hand, it has been shown that the induction of AOX by a respiratory-inhibiting fungicide produces a threefold increase in AOX gene transcription [48] and that *Bcaox* is associated with an alternative response mechanism against oxidative stress [53], participates in various metabolic processes essential for the survival of *B. cinerea*, and it is involved in the virulence of this fungi [54]. Thus, we have considered it interesting to assess the gene expression of *Bcaox*, *Cas-1*, and *Bchex* in the absence and presence of the most active compounds, i.e., (**9**) and (**13**). The *Cas-1* gene is a metacaspase, which belongs to the high-temperature requirement (HtrA) family of serine proteases, and is a homolog of the human HtrA2/Omi, a mitochondrial protein with proapoptotic activity [55], whereas *Bchex* increases their expression with hypha-damaging effects because Woronin bodies appear to plug the septal pores within a few minutes [56,57]. Analysis of the gene expression was performed using RNA extracted from conidia at the early stages of germination instead of hyphae. The reason for this choice is that spores are an isolated and simpler structure, and the inhibition mechanism should be the same. Changes in gene expression were analyzed using the qRT-PCR (quantitative real-time PCR) technique, which allows for the determination of changes in the expression of the genes of interest, relative to the expression of an endogenous gene. Fold changes in transcript accumulation in treated samples relative to NC were calculated using the 2^−ΔΔCT^ method [58].

The results shown in Figure 4 show that both compounds (**9**) and (**13**) downregulate the expression of *Bcaox* at early stages in the germination of *B. cinerea* conidia. This is an interesting result because there is evidence that the induction of AOX by a respiratory-inhibiting fungicide produces a threefold increase in AOX gene transcription [48]. Therefore, these results indicate that 2-allylphenol derivatives affect *B. cinerea* at two different stages of development, i.e., they inhibit the respiratory system of mycelia and downregulate the *Bcaox* expression of conidia. The relationship between AOX inhibition and the control of pathogenic fungi is well established and, therefore, much effort has been dedicated to the development of new fungicides targeting it [59,60]. Interestingly, it has been suggested that flavonoid compounds existing in infected plants can suppress the AOX pathway [61]. Thus, a synergic effect on respiratory inhibition could be obtained by the association of respiratory and AOX inhibitors. However, highly effective and specific AOX inhibitors are currently lacking, so this synergistic effect has not yet been verified in vivo [60].

On the other hand, both derivatives induce a nearly tenfold overexpression of *Bchex*, which could be related to the stress suffered by conidia exposed to these inhibitors. The inhibition of AOX activity might cause an increase in ROS production, which finally induces death by oxidative stress. Survival attempts of fungi involve the increase in Woronin bodies to seal the septa. Finally, no statistically significant *Cas-1* expression changes are observed in the presence of these compounds, which implies that the inhibition of germ tube development is not related to an apoptotic process [30].

The results obtained in this work suggest that 2-allylphenol and its derivatives are a promissory platform for the development of this kind of antifungal agent acting on the respiratory system of *B. cinerea*.

## 3. Methods and Materials

### 3.1. General Information

2-allylphenol (**3**) purchased from Aldrich (St. Louis, MO, USA) was of the highest commercially available purity and was used without additional purification. 2-allylphenol derivatives were synthesized following previously described procedures [28], except for compound (**11**). The synthesis and characterization of this compound are described below. IR spectra were obtained in a Thermo Scientific Nicolet Impact 6700 FT-IR spectrometer (San Jose, CA, USA) using KBr pellets, and the frequencies were reported in cm^−1^. ^1^H and ^13^C-NMR (DEPT 135 and DEPT 90) were performed on a Bruker Avance NEO 400 Digital NMR spectrometer (Bruker, Rheinstetten, Germany), operating at 400.1 MHz for ^1^H and 100.6 MHz for ^13^C. Spectra were recorded in CDCl_3_ solutions and were referenced to the residual peaks of CHCl_3_, δ = 7.26 ppm and δ = 77.0 ppm for ^1^H and ^13^C, respectively. Chemical shifts were reported in δ ppm and coupling constants (*J*) were given in Hz. GC–MS was carried out using a SHIMADZU GCMS-QP2010 instrument (Tokyo, Japan) using a 30 m × 0.25 mm id., 0.25 μm Rtx-5MS capillary column with helium as the carrier gas at a flowrate of 1.61 mL/min. The column temperature was 60 °C for 1 min at 5 °C/min, then increased to 285 °C for 2 min at 15 °C/min. The purification of the reaction products was performed by flash chromatography, using a GILSON PLC 2250 instrument (Saint-Avé, France) with silica cartridges, 40–63 µm, 60 Å (SiliaSep Flash Cartridges, SiliCycle Inc., Quebec, QC, Canada), and thin layer chromatography (TLC). TLC spots on silica gel plates GF-254 (Aldrich, St. Louis, MO, USA) were detected by UV light and heating after spraying with 25% H_2_SO_4_ in H_2_O.

### 3.2. Chemistry

#### Synthesis of 1-Allyl-2-methoxy-5-nitrobenzene (**11**)

2-allyl-4-nitrophenol (**7**) 0.5 g (2.23 mmol) was dissolved in acetone (50 mL) and subsequently, 2.5 g of calcium carbonate (Aldrich, St. Louis, MO, USA) (18.1 mmol) and 2 mL of dimethyl sulfate (Aldrich, St. Louis, MO, USA) (21 mmol) were added. The reaction was left to continue overnight under reflux. After this period, the complete disappearance of (**7**) was confirmed by TLC (ethyl acetate/hexane; 1:3 by volume). The reaction product was diluted in acetone (30 mL) and water (50 mL), and then extracted with dichloromethane (3 × 50 mL), dried with anhydrous sodium sulfate, filtered, and vacuum evaporated. Pure derivative (**11**) (0.48 g; 89.0% of yield) was obtained by flash chromatography employing mixtures of increasing polarity of hexane/ethyl acetate.

Compound (**11**): IR (KBr, cm^−1^): 3085 (=C-H); 2937 (C-H); 1602 (C=C); 1523 (NO_2_); and 1353 (N=O). ^1^H-NMR: 7.63 (1H, d, *J* = 1.6 Hz, H-5); 7.43 (1H, d, *J* = 1.6 Hz, H-6); 7.15 (1H, dd, *J* = 1.4 and 7.8 Hz, H-3); 6.00–5.90 (2H, m, H-2′); 5.15–5.06 (2H, m, H-3); 3.88 (3H, s, CH_3_CO); and 3.48 (2H, d, *J* = 7.7 Hz, H-1). ^13^C-NMR: 151.4 (C-2); 144.2 (C-3); 136.4 (C-1); 135.7 (C-2′); 135.0 (C-6); 123.8 (C-5); 123.4 (C-4); 117.0 (C-3′); 62.6 (OCH_3_); and 33.5 (C-1′). EI–MS (+) *m*/*z* 193 [M+] (100%). Spectra’s ^1^H-NMR (Appendix A), ^13^C-NMR (Appendix A) and EI–MS (Appendix A) were supported in Appendix A.

### 3.3. Biological

#### 3.3.1. Fungal Isolate and Culture Conditions

The PN2 isolate of *B. cinerea* used in this study was maintained and grown under conditions previously described [24].

Conidia (1 × 10^6^ spores/mL) were incubated in a potato dextrose medium for 5–7 days at 21 °C.

#### 3.3.2. Effect of 2-Allylphenol Derivatives on Mycelial Growth

Antifungal activities of 2-allylphenol and its derivatives were evaluated using the radial growth test on malt-yeast extract agar [30]. All tested compounds were dissolved in dichloromethane at different concentrations. Then, 200 μL of each solution were added to 7 mL of malt-yeast extract agar. Controls were prepared in the same way using pure solvent instead of tested compound dissolutions. Control and compounds (**3**)–(**15**) were poured into 6 cm diameter Petri dishes, which were left open in a biosecurity hood for 40 min to remove the solvent. After solvent evaporation, the Petri dishes were inoculated with a 5 mm diameter agar disc containing the mycelium of the PN2 isolate of *B. cinerea*. After two days of incubation in the dark at 22 °C, the colony diameter was measured and growth inhibition percentages were calculated using Equation (1)
(1)$$\mathrm{Growth}\;\mathrm{inhibition}\;(\%I)=\frac{\left(d_{C}-d_{0})-(d_{S}-d_{0}\right)}{\left(d_{C}-d_{0}\right)}\times 100$$
where *d*_0_, *d_C_*, and *d_S_* represent the diameter (in mm) of the fungus agar plug and the colony diameter in control and compound-containing dishes, respectively. The inhibition percentages were plotted as a function of tested compound concentration and fitted to a dose-response equation. These fittings provided the IC_50_ values listed in Table 1. Plotting and fitting of the data and obtention of IC_50_ values were carried out using Origin v8.0 software (OriginLab, Northhampton, MA, USA). Significant differences were evaluated with a two-way analysis of variance (Tukey’s test; *p* < 0.05).

### 3.4. Effect on Gene Expression

#### RNA Extraction and qRT-PCR Analysis

RNA extraction and qRT-PCR analysis were performed according to a previously reported method [62]. Briefly, RNA extraction was carried out using the Total RNA extraction kit of MN (Macherey-Nagel, Düren, Germany). Once the quality parameters (1.5% MOPS gel in 1X APR) and RNA quantity (Infinite 200 PRO, Tecan Trading AG, Männedorf, Switzerland) were determined, 1 μg of RNA was used for cDNA synthesis using the GoScript Retro Transcription kit (Promega, WI, USA) following the manufacturer’s instructions. Reactions were performed using Takyon™ One-Step MasterMix (Eurogentec, Seraing, Belgium) in Lightcycler 96 (Roche, Basel, Switzerland) thermal cycler equipment under conditions suggested by the manufacturer. Four primer pairs with the same efficiency were used (Appendix A). Cycle threshold (Ct) values obtained for *UbcE*, as a housekeeping gene, were used to normalize the data [63]. This is an endogenous gene for constitutive expression that is not affected by these treatments. For all qRT-PCR experiments, two independent biological replicates were included, and reactions were performed in triplicate. Fold changes in transcript accumulation in treated samples relative to nontreated samples were calculated using the 2^−ΔΔCt^ method [58].

## 4. Conclusions

A series of 2-allylphenol derivatives have been evaluated for their antifungal activity on the mycelial growth of *B. cinerea*. Our results indicate that the replacement of the hydroxyl group by methoxy or acetyl groups produces a strong increase in activity, i.e., IC_50_ values diminish from 68.0 to 2.00 and 1.00 μg mL^−1^, respectively. As these structural modifications of (**3**) are minor, we assume that these derivatives follow the same mechanism of action, i.e., the inhibition of respiration by blocking electron transport in the cytochrome pathway. Thus, the huge changes in activity observed for derivatives (**9**) and (**11**) have been attributed to competitive binding to the same receptors of the couple ubiquinol/ubiquinone. Additionally, it was found that the most active derivatives ((**9**) and (**11**)) downregulate the expression of *Bcaox* in the early stages of *B. cinerea* conidia germination. This gene is associated with the activation of alternative oxidase enzyme (AOX), which limits the efficiency of respiratory inhibitors by providing an alternative pathway by which the fungus respiration can continue, allowing the ATP level necessary for its metabolic activity to be recovered.

Therefore, we believe that these compounds are a very attractive scaffold for the development of antifungal agents against *B. cinerea*.

## Figures and Tables

**Figure 1 ijms-24-06530-f001:**
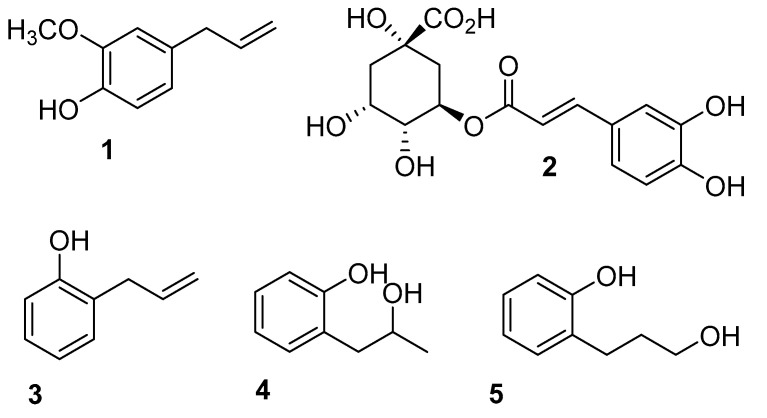
Chemical structure of natural phenylpropanoids: Eugenol (**1**), chlorogenic acid (**2**), 2-allylphenol (**3**), 2-(2-hydroxypropyl) phenol (**4**), and 2-(3-hydroxypropyl) phenol (**5**).

**Figure 2 ijms-24-06530-f002:**
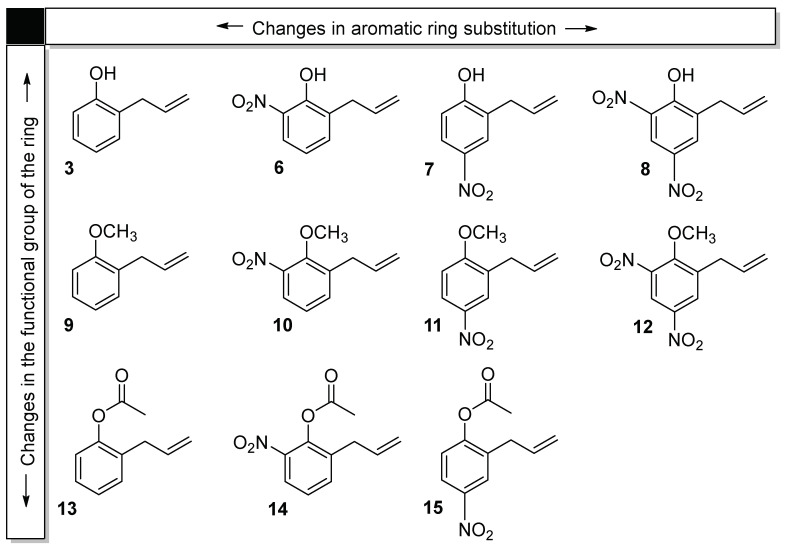
Chemical structure of 2-allylphenol (**3**) and its derivatives (**6**–**15**), used in this study.

**Figure 3 ijms-24-06530-f003:**
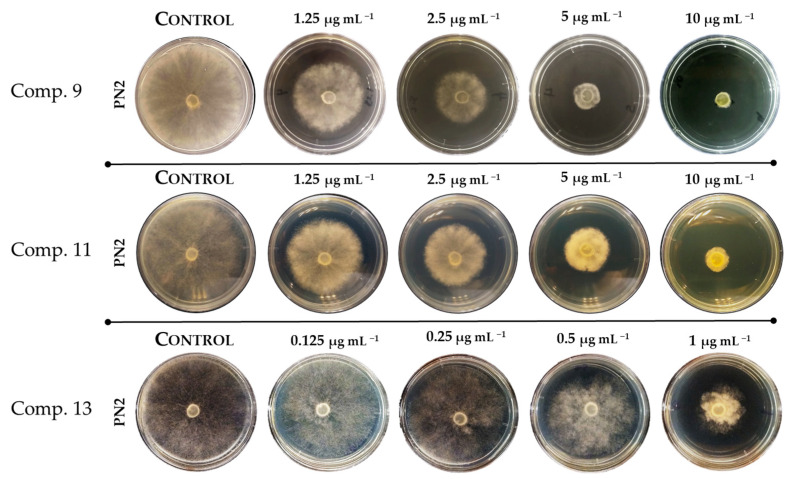
Effect of 2-allylphenol derivatives on the mycelial growth of *B. cinerea* in solid PDA medium (Difco Potato Dextrose Agar). These images show the effect on hyphae growth from the multiresistant strain of *B. cinerea* (PN2) in the presence of different concentrations of compounds (**9**), (**11**)**,** and (**13**).

**Figure 4 ijms-24-06530-f004:**
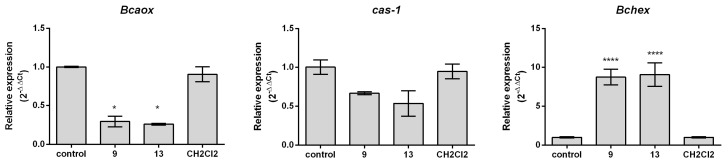
The transcript expression of genes *Bcaox*, *Cas-1*, and *Bchex* in the spores of *B. cinerea* after the application of derivatives (**9**) and (**13**). The relative transcript expressions were analyzed by real-time PCR. The unity value of relative expression corresponds to nontreated samples. All data represent the means of three independent replicates ± SD (*n* = 9). The significance of the difference was analyzed by Sidak’s multiple comparison tests (* *p* < 0.05); asterisks indicate significant differences compared with negative control under the same treatment conditions (**** *p* < 0.0001).

**Table 1 ijms-24-06530-t001:** IC_50_ values for the in vitro inhibition of the mycelial growth of *B. cinerea* obtained for 2-allylphenol and its derivatives. These values were estimated by measuring mycelium growth after 48 h of incubation.

2-Allylphenol Derivatives	PN2
	IC_50_ (μg mL^−1^)
3	68.0 ± 6.40
6	133 ± 12.8
7	13.0 ± 1.50
8	136 ± 11.5
9	2.00 ± 0.32
10	55.0 ± 4.85
11	3.60 ± 0.28
12	12.0 ± 1.52
13	1.00 ± 0.15
14	17.5 ± 2.30
15	16.0 ± 1.50

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
