# Peer review of "Antifungal Activity of 2-Allylphenol Derivatives on the *Botrytis cinerea* Strain: Assessment of Possible Action Mechanism"

_ijms, 2023, doi:10.3390/ijms24076530_

Round 1

Reviewer 1 Report

Dear Corresponding Author

For evaluation of fungicidal activity of a compound it is really required to examine it as in vivo on a model plant (for instance., Arabidopsis thaliana).

Regards

Author Response

Reviewer 1

Dear Corresponding Author

For evaluation of fungicidal activity of a compound it is really required to examine it as in vivo on a model plant (for instance., Arabidopsis thaliana).

We agree with this reviewer that the final assay of fungicidal activity must be carried out in vivo. However, it is widely accepted that experiments in vitro are the first stage for getting a screening of fungicidal activity as a function of chemical structure, and a plausible mechanism. In this work we are measuring fungicidal activity for a series of derivatives of 2-allylphenol, and experiments in vivo will be performed using those compounds presenting the highest activity.

Reviewer 2 Report

The manuscript ID ijms-2293002 describes the synthesis of 2-allylphenol derivatives and their activity against Botritys cinerea mycelial growth and the expression of three genes of conidia. The manuscript is interesting and includes relevant information for readers. However, several points should be addressed before being considered further.

1.      Detailed scrutiny should be performed throughout the manuscript to look for some grammar, stylistic, and even typos issues.

2.      Line 21: The inhibitory concentrations should be expressed in micromolar or micrograms/mL instead of ppm. Be consistent throughout the manuscript. In addition, the units must be unified.

3.      Line 19-21: This sentence should be revised since it is confusing. In addition, are these IC50s related to mycelial growth or spore germination?

4.      Line 67; Specify the active principle of BC-1000. Is it related to the allylphenols? What is the foundation of such a comparison?

5.      Line 92: …relative para position…to what? Allyl or hydroxyl group? Specify…

6.      Table 1: Using the significant figures of IC50 values is incorrect and should be revised. In addition, the confidence interval of the non-linear regression to determine the IC50s must be informed. Finally, statistics over these IC50 values must be performed.

7.      Line 129: 68 versus. 48.8 are strictly similar. They have a 28.2% relative difference, so I recommend revising this passage to avoid confusing interpretations.

8.      Lines 155-163: These sentences should be revised since they are highly confusing.

9.      Line 163: In the abstract, there is a mention of activity on spore germination, but no related activity is developed in the manuscript. It just mentioned using the early conidia stage on the transcriptional levels, but no effect on spore germination was investigated. So, such a mention of spore germination must be removed from the manuscript.

10.   Line 195: Why measure RNA levels of genes in conidia if the experimental effect was determined on mycelial growth? There is an inconsistency there that must be clarified in the manuscript.

11.   Figure 3: How was the procedure to measure the mycelial growth diameters in the Petri dishes? This information is not specified in section 3.3.2. In addition, how was the mycelium aerially growing interpreted as can be observed in compound 13 at 0.05 ppm?

12.   Line 251: Spectral plot of compound 11 must be provided as supplementary material. In addition, HRMS data is missing and should be included.

13.   Line 277: The confidence interval is missing and must be informed.

14.   Lines 278 and 298: It is mentioned that ANOVA was used as inferential statistics, but such a piece of information is not used in describing or discussing the results. In addition, why was ANOVA used if a normal data distribution was not verified? Such verification of data normality must be included.

15.   Line 282: The details on conidia culture are missing and must be provided.

16.   Line 290: The sequences of the primers of target and housekeeping genes must be provided.

17.   The R&D section is more oriented toward describing results. Therefore, a more comprehensive discussion is missing and should be improved.

18.   The conclusion section summarizes the results, but conceptual findings from the mechanistic point of view should be provided.

19.   Line 315: why platform? I consider this term incorrectly used. I suggest considering the term scaffolds, lead compounds or head series, or active principles, depending on the purpose intended by the authors for this term, which is highly confusing. Be consistent throughout the manuscript.

Author Response

Reviewer 2

The manuscript ID ijms-2293002 describes the synthesis of 2-allylphenol derivatives and their activity against Botritys cinerea mycelial growth and the expression of three genes of conidia. The manuscript is interesting and includes relevant information for readers. However, several points should be addressed before being considered further.

  1. Detailed scrutiny should be performed throughout the manuscript to look for some grammar, stylistic, and even typos issues.

It has been done

  1. Line 21: The inhibitory concentrations should be expressed in micromolar or micrograms/mL instead of ppm. Be consistent throughout the manuscript. In addition, the units must be unified.

The inhibitory concentrations have been expressed in μg mL-1 throughout the manuscript.

  1. Line 19-21: This sentence should be revised since it is confusing. In addition, are these IC50s related to mycelial growth or spore germination?

This sentence has been modified and now it reads “For example, changing the hydroxyl group by methoxy or acetyl groups produces dramatic increases on mycelial growth inhibition, i.e. IC50 value of 2-allylphenol decreases from 68 to 2 and 1 μg mL-1, respectively”

  1. Line 67; Specify the active principle of BC-1000. Is it related to the allylphenols? What is the foundation of such a comparison?

BC-1000 is a natural fungicide obtained from grapefruit that is widely used in Chile and other countries of South America for control of B. cinerea. According to the supplier its activity could be attributed to bioflavonoids. Here and in previous work we have used it as control to compare antifungal activities. In any case, we have modified the sentence in line 67 and now it reads “…like that of BC-1000 (Chemie SA), a natural fungicide used for control of B. cinerea.”

  1. Line 92: …relative para position…to what? Allyl or hydroxyl group? Specify…

We use ortho and para to denote the position of nitro groups relative to hydroxyl, methoxy, or acetyl group in the molecule. In this case, to make it clearer we have rewritten the phrase starting in line 90. Now it reads “Interestingly, the antifungal activity is enhanced in those compounds where a nitro group is attached in para position relative to the hydroxyl group”.

  1. Table 1: Using the significant figures of IC50values is incorrect and should be revised. In addition, the confidence interval of the non-linear regression to determine the IC50s must be informed. Finally, statistics over these IC50 values must be performed.

In table 1 we have added the standard deviation for IC50 values given by the fitting of data to the dose-response curve. The average value and its standard deviation are given automatically by the program that consider the three replicas at the same time.

  1. Line 129: 68 versus. 48.8 are strictly similar. They have a 28.2% relative difference, so I recommend revising this passage to avoid confusing interpretations.

This phrase was rewritten and now it reads “Results listed in Table 1 indicate that IC50 varies from 1 to 136 μg mL-1, and the value obtained for 2-allylphenol (68 μg mL-1) is in the same range of that previously reported (48.8 μg mL-1) for inhibition of B05.10, a non-resistant strain of B. cinerea”

  1. Lines 155-163: These sentences should be revised since they are highly confusing.

The paragraph in lines 156-166 was completely rewritten and we hope that its meaning has been clarified.

  1. Line 163: In the abstract, there is a mention of activity on spore germination, but no related activity is developed in the manuscript. It just mentioned using the early conidia stage on the transcriptional levels, but no effect on spore germination was investigated. So, such a mention of spore germination must be removed from the manuscript.

We have eliminated the mention of spore germination on the abstract

  1. Line 195: Why measure RNA levels of genes in conidia if the experimental effect was determined on mycelial growth? There is an inconsistency there that must be clarified in the manuscript.

Analysis of gene expression were carried out on conidia because they are isolated and more simple structures, whose response to external factors can be more easily interpreted and discussed. This has been added to the manuscript in lines 188-191

  1. Figure 3: How was the procedure to measure the mycelial growth diameters in the Petri dishes? This information is not specified in section 3.3.2. In addition, how was the mycelium aerially growing interpreted as can be observed in compound 13 at 0.05 ppm?

Mycelial growth diameters were measured in a daily basis using a ruler and the procedure shown in the following figure

The diameter is obtained as the average of values obtained in the three directions.

  1. Line 251: Spectral plot of compound 11 must be provided as supplementary material. In addition, HRMS data is missing and should be included. (Yo)

1H and 13C RMN spectra, as well as MS spectra are provided as supplementary material.

  1. Line 277: The confidence interval is missing and must be informed.

This was our mistake because we never made a statistical analysis of this kind. We have just reported the average values of IC50 as given by the software that fit data to dose -response curves. Thus, we have eliminated the phrase “Significant differences were evaluated with a two-way analysis of variance”

  1. Lines 278 and 298: It is mentioned that ANOVA was used as inferential statistics, but such a piece of information is not used in describing or discussing the results. In addition, why was ANOVA used if a normal data distribution was not verified? Such verification of data normality must be included.

Similar to answer in Line 277

  1. Line 282: The details on conidia culture are missing and must be provided.

Details of conidia culture have been added in Lines 269-270

  1. Line 290: The sequences of the primers of target and housekeeping genes must be provided.

The identification and structure of primers are given in Table S1. A sentence mentioning this has been included in line 298

  1. The R&D section is more oriented toward describing results. Therefore, a more comprehensive discussion is missing and should be improved.

We believe that the main results are already discussed. However, in lines 155-167 we have improved the discussion regarding the changes on activity due to substitution in the aromatic ring. The changes are explained in terms of variations on binding ability to the receptor involved in the cytochrome pathway.

  1. The conclusion section summarizes the results, but conceptual findings from the mechanistic point of view should be provided.

A paragraph has been added in Lines 316-318 to emphasize the mechanism of action of the most active derivatives

  1. Line 315: why platform? I consider this term incorrectly used. I suggest considering the term scaffolds, lead compounds or head series, or active principles, depending on the purpose intended by the authors for this term, which is highly confusing. Be consistent throughout the manuscript.

We have changed the word “platform” by “scaffold”

Round 2

Reviewer 1 Report

Dear Corresponding Author

With your corrections I think the paper is now suitable for publication.

Regards

Reviewer 2 Report

The authors adequately addressed my comments, so the manuscript can be accepted for publication.